# Two-Dimensional TeB Structures with Anisotropic Carrier Mobility and Tunable Bandgap

**DOI:** 10.3390/molecules26216404

**Published:** 2021-10-23

**Authors:** Yukai Zhang, Xin Qu, Lihua Yang, Xin Zhong, Dandan Wang, Jian Wang, Baiyang Sun, Chang Liu, Jian Lv, Jinghai Yang

**Affiliations:** 1Key Laboratory of Functional Materials Physics and Chemistry of the Ministry of Education, College of Physics, Jilin Normal University, Changchun 130103, China; Lance19950815@163.com (Y.Z.); zhongxin19881119@163.com (X.Z.); mila880227@126.com (D.W.); wangjianphy@163.com (J.W.); ng68733384@163.com (B.S.); liuchang8081858@163.com (C.L.); jhyang1@jlnu.edu.cn (J.Y.); 2Key Laboratory of Preparation and Application of Environmental Friendly Materials, College of Physics, Jilin Normal University, Changchun 130103, China; 3International Center for Computational Method and Software, State Key Laboratory of Superhard Materials, College of Physics, Jilin University, Changchun 130012, China; lvjian@jlu.edu.cn; 4Key Laboratory of Physics and Technology for Advanced Batteries (Ministry of Education), College of Physics, Jilin University, Changchun 130012, China

**Keywords:** two-dimensional TeB structures, carrier mobility, horizonal or lateral heterostructures, density functional theory

## Abstract

Two-dimensional (2D) semiconductors with desirable bandgaps and high carrier mobility have great potential in electronic and optoelectronic applications. In this work, we proposed *α*-TeB and *β*-TeB monolayers using density functional theory (DFT) combined with the particle swarm-intelligent global structure search method. The high dynamical and thermal stabilities of two TeB structures indicate high feasibility for experimental synthesis. The electronic structure calculations show that the two structures are indirect bandgap semiconductors with bandgaps of 2.3 and 2.1 eV, respectively. The hole mobility of the *β*-TeB sheet is up to 6.90 × 10^2^ cm^2^ V^−1^ s^−1^. By reconstructing the two structures, we identified two new horizontal and lateral heterostructures, and the lateral heterostructure presents a direct band gap, indicating more probable applications could be further explored for TeB sheets.

## 1. Introduction

Two-dimensional (2D) layered materials are crystalline materials made of a few layers of atoms with nanometer or less thickness. Two-dimensional layered materials, such as graphene [1], boron-nitride [2], phosphorene [3], transition metal dichalcogenides [4], and Mxene [5] have attracted considerable attention in recent years due to their unique electronic [6], mechanical [7], and optical properties [8,9]. Due to these excellent properties of 2D materials, they also have important applications in materials science [10], energy [11,12], biomedicine [13], and drug delivery [14], and are considered as a revolutionary material in the future. Therefore, it is crucial to discover and design new kinds of novel 2D materials.

Two-dimensional layered boron-based materials, as the key members of the 2D materials family, have many unique properties and intriguing applications. Borophenes with different structures have been successfully synthesized experimentally recently [15,16]. Borophenes exhibit structural anisotropies and polymorphisms, which result in a range of properties unique to 2D boron-based materials including a combination of mechanical flexibility, metallicity, transparency, superconductivity, etc. [17]. Two-dimensional borides represent a large part of 2D boron-based materials, their varied properties have been extensively investigated. B_4_N possesses superior mechanical flexibility in ideal tensile strength and critical strain values and is also predicted to have a negative Poisson’s ratio; these properties show it can be used in future nanomechanical devices [18]. AlB_6_ is predicted to possess high mechanical stability, the in-plane Young’s modulus is larger than that of graphene, suggesting that AlB_6_ has remarkable mechanical properties [19]. Two-dimensional borides materials also have great potential as anode materials in metal-ion batteries. Zr_2_B_2_ has also been shown as a promising two-dimensional anode material for Li-ion batteries—the theoretical specific capacity of 526 mAh g^−1^ is larger than that of the commercial graphite electrode [20]. Monolayer B_7_P_2_ as an anode material for lithium/sodium-ion batteries has a high storage capacity of 3117 mA h g^−1^ for both lithium/sodium-ion batteries [21]. Two-dimensional borides materials have been shown as excellent superconducting materials. B_2_O is an intrinsic superconductor, superconducting transition temperature (*T*_c_) of ~10.3 K [22]. AlB_6_ is a superconducting material and its *T*_c_ can be greatly enhanced up to 30 K by applying tensile strain at 12%, which shows great potential in the field of superconductivity [19]. Meanwhile, X_2_B_2_ (X = Li, Mo, W) [6,23,24] and other XB_6_ (M = Ga, In, Mg, Ca, Ti, Y, Sc) [19,25,26] are also predicted to have superconductivity. In recent years, more and more 2D boron layers have been theoretically predicted with planar hypercoordinated motifs [27,28,29]. Theoretical calculations also suggest a lot of applications for 2D borides, such as in magnetic devices [30], catalysis [27], electroreduction [31]. The excellent properties and varied potential applications make the exploration of 2D borides significant.

As an important member of chalcogenides, 2D telluride atomic crystals have been intensely investigated because of their unique electronic structures and diverse 2D atomic crystals [32]. GeTe monolayer is a semiconducting material with a considerable band gap of 2.35 eV and shows suitable band edge positions for photocatalytic water splitting [33]. Both PbTe and GeTe monolayer exhibit rather high carrier mobilities, while GeTe has pronounced optical absorption in the visible region of the solar spectrum [34]. The WTe_2_ [35] exhibits high carrier mobility (2100 cm^2^ V^−1^ s^−1^), suggesting the application prospect of atom-thick telluride atomic crystals in high-speed logic transistors [36] and supercapacitor [37]. Furthermore, Te-based 2D ferromagnetic and superconducting materials have recently made some significant breakthroughs such as gate-tunable Fe_3_GeTe_2_ [38] and Cr_2_Ge_2_Te_6_ [39] vdWs ferromagnetic semiconductors and electrically-tunable monolayer WTe_2_ superconductor [40]. Furthermore, Te-based 2D topological materials are remarkably noteworthy including topological insulators (Bi_2_Te_3_ and Sb_2_Te_3_), topological crystalline insulators (SnTe) and Weyl and Dirac semimetals (T_d_-MoTe_2_ and T_d_-WTe_2_) [41].

In this work, we predicted two novel TeB monolayers noted as *α*-TeB and *β*-TeB, by using density functional theory(DFT) combined with the particle swarm-intelligent global structure search method. These two TeB monolayers are confirmed to have thermal and dynamical stability which means that they may probably be synthesized experimentally. Moreover, the high anisotopic carrier mobility of both TeB monolayers indicates that *α*-TeB and *β*-TeB might be suitable for novel nanoelectronic applications with high on-off ratios. By stacking and jointing *α*-TeB and *β*-TeB, we reconstruct horizontal and lateral heterostructures. The band gap value becomes smaller in the low energy horizontal heterostructure and the direct band gap semiconductor appeared at a low energy lateral heterostructure. Our work enriches the large family of 2D compounds and also provides new materials for the follow-up study of 2D boride materials in nanoelectronic applications.

## 2. Computational Details

An unbiased swarm-intelligence structural method, as implemented in the CALYPSO code, [42] is employed to explore stable two-dimensional tellurium borides. Its effectiveness has been validated by reproducing already known materials, including either elemental solids and binary/ternary compounds as well [43,44,45]. Moreover, many of the new materials predicted with CALYPSO have been experimentally confirmed [46,47]. Notably, theoretical calculations play an important role in discovering new 2D materials and understanding the physical mechanism implied by experimental phenomena [28,48,49,50,51,52].

Structural optimization and property calculations are performed within the density functional theory framework, as implemented in the Vienna ab initio simulation package (VASP) [53]. The electronic exchange-correlation functional is treated using the generalized gradient approximation (GGA) in the form proposed by Perdew, Burke, and Ernzerhof (PBE) [54]. Atomic positions are fully relaxed until the force on each atom is less than 10^−3^ eV/Å. The supercell method is considered to simulate the monolayer, where a vacuum distance of ~20 Å is used to eliminate the interaction between adjacent layers. Considering that the GGA usually underestimates the band gaps, we adopt the Heyd−Scuseria−Ernzerhof (HSE06) hybrid functional to calculate the band structures [55]. Dynamic stabilities and phonon dispersion curves are computed with the supercell approach as implemented in the Phonopy code [56]. On the other hand, in the molecular dynamics (MD) simulations, the initial configuration in the supercell is annealed at different temperatures, each MD simulation in the NVT ensemble lasts for 10 ps with a time step of 2.0 fs, and the temperature is controlled by using the Nose−Hoover method [57].

## 3. Results and Discussion

### 3.1. Geometric Structures and Stability of α-TeB and β-TeB

Through extensive structure searches, we found two new TeB monolayer stable structures, noted *α*-TeB and *β*-TeB. The predicted monolayers are shown in Figure 1a,b the lattice information is summarized in Table 1. The basic building block of the two structures are all Te_2_B_2_ units; these two TeB sheets present the hexagonal structures, it is clear that these two TeB structures consist of four-atom layers in the order Te-B-B-Te, in other words, both these two TeB structures consist of two pucker honeycomb Te-B atoms’ layers. For the *α*-TeB structure, as can be seen in Figure 1a, it can be seen that it is formed by two Te-B atom layers dislocation stacking or AB stacking, which means the every Te atom on the lower site are located in the hole site in the upper Te-B atom layer, while the B atoms located at same position both in upper and lower site. As for the *β*-TeB structure, as we can see in Figure 1b, it is clearly shown that this structure is similar to the *α*-TeB in the whole configuration—they are a hexagonal construction. However, the *β*-TeB structure is different from the *α*-TeB structure in form details, both the Te and B atoms are located at the same position both in the upper and lower sites, which can be realized as formed by two pucker honeycomb Te-B atoms’ layers with AA stacking. In our DFT calculation, the *α*-TeB is the global minimum structure, *β*-TeB is the local minimum structure which the energy is relatively 46 meV/atom higher than the *α*-TeB. In the *α*-TeB, the lattice parameter is a = b = 3.598 Å, the average B-B bond length is 1.68 Å, and the Te-B bond length is 2.32 Å. While for the *β*-TeB, the lattice parameter is a = b = 3.565 Å, the average B-B bond length is 1.71 Å, and the Te-B bond length is 2.31 Å. The thickness of these two phases is also similar, *α*-TeB has a thickness of 3.632 Å and *β*-TeB has a thickness of 3.837 Å.

We later investigate the electron localization function (ELF), which is used to analyze the bonding character. Generally, the large ELF value (>0.5) corresponds to a covalent bond or core electrons, whereas the ionic bond is represented by a smaller ELF value (<0.5). An ELF value of 0.5 is the metallic bond [58]. The analysis of the electron localization function clearly shows that the Te-B and B-B bonds are all covalent bonds, the bond type also confirms the stability of the two structures. The Bader charge analysis [59] reveals that each Te atom donates its 0.11 *e* to a B atom for *α*-TeB, while each Te atom donates its 0.12 *e* to a B atom for the*β*-TeB, indicating the mostly covalent character of the Te-B bonds of the two Te-B structures. As a consequence, the existence of covalent bonds leads to good structural stability.

Cohesive energy is a well-accepted parameter to evaluate the feasibility for experimental synthesis of the predicted two-dimensional materials [60,61]. Here, the cohesive energy (E_coh_) is performed by:(1)$$E_{\mathrm{coh}}\;\frac{\left(E_{\mathrm{Te}}+E_{B}-E_{\mathrm{TeB}}\right)}{2}$$
where E_TeB_ is the total energy of primitive cell of TeB, and E_Te_ and E_B_ are the energies of isolated Te and B atoms, respectively. The obtained E_coh_ is calculated to be 4.22 eV for *α*-TeB and 4.18 eV for *β*-TeB, respectively, they are higher than B_2_O (2.43 eV/atom) [22] and arsenene allotropes(3.04–3.20 eV/atom) [62], indicating the exothermic process and experimental feasibility under suitable external conditions.

We further analyze the stability of the *α*-TeB and *β*-TeB monolayers by calculating the phonon dispersion relation. The phonon band structure results are presented in Figure 2a,b. The absence of imaginary frequencies in the phonon band structures confirmed the dynamical stability of *α*-TeB and *β*-TeB monolayers. In particular, the highest frequency of *α*-TeB structure (981.02 cm^−1^, 29.46 THz) and *β*-TeB structure (939.39 cm^−1^, 28.21 THz) is even higher than those of 473 cm^−1^ in the MoS_2_ monolayer [63], 580 cm^−1^ in silicene [58], and 854 cm^−1^ in the FeB_2_ Monolayer [60], indicating the robust connection between the Te and B atoms in the two structures.

Additionally, another key factor to check structure stability is its thermal stability at a high temperature, we next study their thermal stability by performing AIMD simulations. We adopt a 3 × 3 × 1 supercell to minimize the effects of periodic boundary conditions. The results of the variation of free energy in the AIMD simulations within 10 ps, along with the last frame of the photographs, are exhibited in Figure 2c,d (more details of AIMD simulations are shown in Appendix A). The two structures *α*-TeB and *β*-TeB maintain structural integrity up to 1000 K and 700 K, respectively. It is clear that the thermodynamic stability of the *α*-TeB structure is higher than that of hex-GaB_6_ and rect-InB_6_ are both stable at 500 K, and hex-InB_6_ only retains integrity under 300 K, MoB_2_ is stable at 500 K [64]. The well-maintained geometries at 700 K or 1000 K indicate that *α*-TeB and *β*-TeB can not only be applied at room temperatures but might also have possible applications at a relatively high temperature.

### 3.2. Mechanical Properties, Electronic Properties and Anisotropic Carrier Mobility

We subsequently study the mechanical properties. Then, Young’s moduli and Poisson’s ratio can be obtained with the aid of the VASPKIT [65], a post-processing program for the VASP code. Our calculations estimate the value of C_11_, C_12_, C_21_, C_22_, C_66_ for *α*-TeB and *β*-TeB are shown in Table 2.

Obviously, the two Te-B monolayers satisfied the Born criteria [66], C_11_ > 0, C_66_ > 0, and C_11_C_22_–C_21_^2^ > 0 [67], which confirms the mechanical stability of the two structures monolayer. The in-plane Young’s moduli (Y) along the x and y directions are obtained with the help of elastic constants as:(2)$$Y_{x}\;\frac{C_{11\;}C_{12}-{{\;C}_{12}}^{2}}{C_{22}}\;\mathrm{and}\;Y_{y}\;\frac{C_{11\;}C_{12}-{{\;C}_{12}}^{2}}{C_{11}}$$

It is clear to see in Table 1 that C_11_ = C_22_, C_12_ = C_21_ for the two structures, so the Y_x_ = Y_y_ = 125.48 N/m, 139.00 N/m, respectively. Apparently, Young’s moduli are comparable with MoS_2_ (123 N/m) [7], phosphorene (24–103 N/m), and silicene (62 N/m) [68]. The Poisson’s ratio reflects the mechanical responses, can be calculated as:(3)$$V_{x}=\frac{C_{12}}{C_{22}}\;\mathrm{and}\;V_{y}=\frac{C_{12}}{C_{11}}$$

V_x_ = V_y_ = 0.153 for *α*-TeB and 0.156 for *β*-TeB, indicating the large isotropy in mechanical properties. We also calculated the values of the ideal strength of α-TeB and β-TeB monolayers, due to the similarity of the two structures in three directions (100, 010, 110), Therefore, the values in these three directions are basically the same, see Appendix A for details.

In order to study the electron properties of our predicted structures, we calculate the band structures as well their corresponding electronic total density of states (DOS) and partial density of states (PDOS), as exhibited in Figure 3. The high-symmetry paths of the two structures are both along Γ–M–K–Γ. These two structures are semiconductors due to no bands across the Fermi level, and PDOS shows the main contribution to the states around the Fermi level arises from B-p and Te-p orbitals. At first sight, the distributions of the band structures for the two structures are also similar, this is because of the similarity of structure and configuration. What is noteworthy is that the two structures are indirect band gap semiconductors, the band gap for *α*-TeB and *β*-TeB are 2.3 eV and 2.1 eV, respectively. We also calculated the band decomposed charge densities for VBM and CBM of α-TeB and *β*-TeB monolayers, respectively—as seen in Appendix A.

Carrier mobility is a dominant feature to determine the conductivity, evaluate the working efficiency and assess the electrical performance of semiconductor materials. In this point, we transform *α*-TeB and *β*-TeB structures to rectangle cells for carrier mobility calculation, as shown in Figure 4a,b, respectively. Carrier mobility depends on the effective mass, which in turn is affected by strain. Using effective mass approximation and deformation potential (DP) theory [69], the mobility of 2D materials can be determined as:(4)$$\mu_{i}=\frac{e\hbar^{3}C_{i}}{k_{B}T_{m_{i}m_{d}\;\;}{\left(E_{i}\right)}^{2}}$$
where ℏ is the Planck constant, K_B_ is the Boltzmann constant, i represents the x- or y-axis, T is the temperature, it is used for mobility calculation—300 K for each case. C_i_ is the above-mentioned elastic module along the x- or y-axis, m_i_ is the effective mass along the transport direction, and m_d_ is the average effective mass $m_{d}=\sqrt{m_{x}m_{y}}$. E_i_ is the deformation potential defined by Ei=Δvi(ΔlΔl0), where ∆V_i_ is the energy change of valence band maximum (VBM) or conduction band minimum (CBM) under strain ∆l/∆l_0_. The Carrier type, deformation potentials, elastic modulus, effective electron masses, and estimated carrier mobilities of the *α*-TeB and *β*-TeB sheets along the x- and y-axis at T = 300 K are summarized in Table 3. We can see that the hole and electron mobilities are strongly in-plane anisotropic. Meanwhile, for the two structures, the hole mobility is higher than the electron mobility, and the hole mobility and electron mobility along the x-axis is larger compared to MoS_2_ (~200 cm^2^ V^−1^ s^−1^) [70]. The relatively high and anisotropic hole and electron mobilities in both *α*-TeB and *β*-TeB sheets mean that these two structures should be suitable for novel nanoelectronic applications with high on-off ratios.

### 3.3. Horizonal and Lateral Heterostructures by Stacking and Jointing α-TeB and β-TeB

As above mentioned, the similar lattice parameter and thickness of *α*-TeB and *β*-TeB encourage us to further investigate the horizontal and lateral heterostructures based on both TeB phases. As to whether some new interesting properties would appear, we constructed six structures (Figure 5) by stacking and jointing *α*-TeB and *β*-TeB. After full relaxation of these six structures, the horizontal heterostructures with the lowest energy are shown in Figure 5a and the lateral heterostructures with the lowest energy are shown in Figure 5d, the corresponding band structures to these two structures are shown in Figure 6, respectively. There are two points worth noting: I) the horizontal heterostructure maintained the indirect band, but its band gap (1.92 eV) became narrow compared with *α*-TeB and *β*-TeB, which is slightly less than that of *α*-TeB structure (2.3 eV) and the *β*-TeB structure (2.1 eV). II) The lateral heterostructure transformed to the direct band, and the band gap decreased significantly (0.75 eV). The results show that stacking and jointing *α*-TeB and *β*-TeB can provide an effective method for subsequent band regulation and enlarge the application field of 2D TeB materials.

## 4. Conclusions

In conclusion, we identify two TeB monolayers by combining the global minimum structure search with first-principles calculations. The thermal, and dynamic stabilities indicate that these monolayers are most probably synthesized by appropriate methods. The calculated electronic structures reveal that *α*-TeB and *β*-TeB structures are all semiconductors. In particular, *α*-TeB and *β*-TeB possess indirect band gaps of 2.3 eV and 2.1 eV, respectively. Meanwhile, the electron mobility and hole mobility can reach 3.77 × 10^2^ cm^2^ V^−1^ s^−1^ and 4.78 × 10^2^ cm^2^ V^−1^ s^−1^ for the *α*-TeB structure, and 4.14 × 10^2^ cm^2^ V^−1^ s^−1^ and 6.89 × 10^2^ cm^2^ V^−1^ s^−1^ for *β*-TeB, suggesting its promising applications in electronics. By stacking and jointing *α*-TeB and *β*-TeB, we reconstruct three horizontal heterostructures and three lateral heterostructures. The band gap value became smaller in the low energy horizontal heterostructure and the direct band gap semiconductor appeared as a low energy lateral heterostructure, which provides an effective method for subsequent band regulation and enlarges the application field of 2D materials.

## Figures and Tables

**Figure 1 molecules-26-06404-f001:**
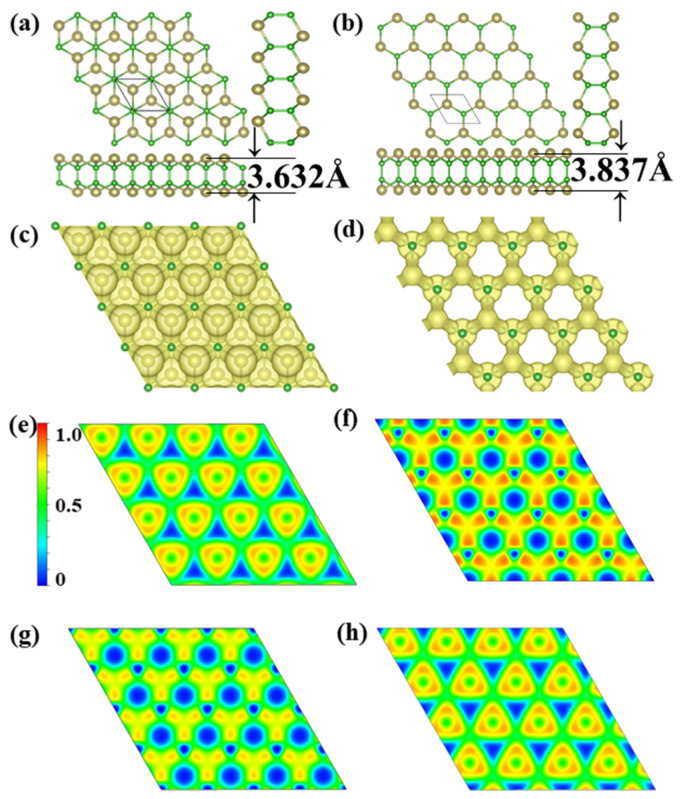
Predicted (**a**) *α*-TeB monolayer with *P*-3*m*1 symmetry and (**b**) *β*-TeB monolayer with *P*-6*m*2 symmetry. The 3D ELF maps of (**c**) *α*-TeB monolayer and (**d**) *β*-TeB monolayer, the isosurfaces of 3D ELFs with the isovalue of 0.75 au. The 2D ELF maps of (**e**) Te-atoms layer and (**g**) B-atoms layer at *α*-TeB monolayer. The 2D ELF maps of (**f**) Te-atoms layer and (**h**) B-atoms layer at *β*-TeB monolayer.

**Figure 2 molecules-26-06404-f002:**
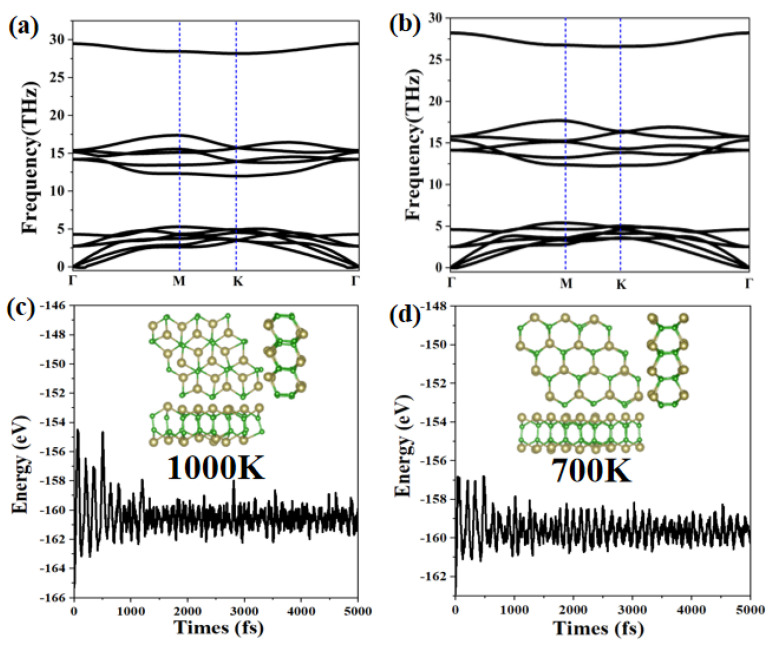
The calculated phonon spectrum for the *α*-TeB (**a**) and *β*-TeB (**b**) monolayers, respectively. The energy fluctuation and the corresponding structure during the AIMD simulations at 1000 K for *α*-TeB (**c**) and 700 K for *β*-TeB (**d**).

**Figure 3 molecules-26-06404-f003:**
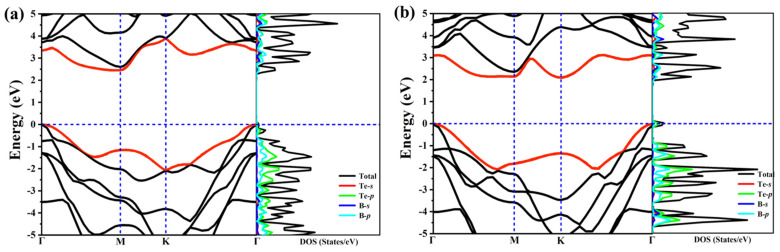
The calculated electronic band structure and PDOS of the *α*-TeB (**a**) and *β*-TeB (**b**) monolayers at the HSE06 level.

**Figure 4 molecules-26-06404-f004:**
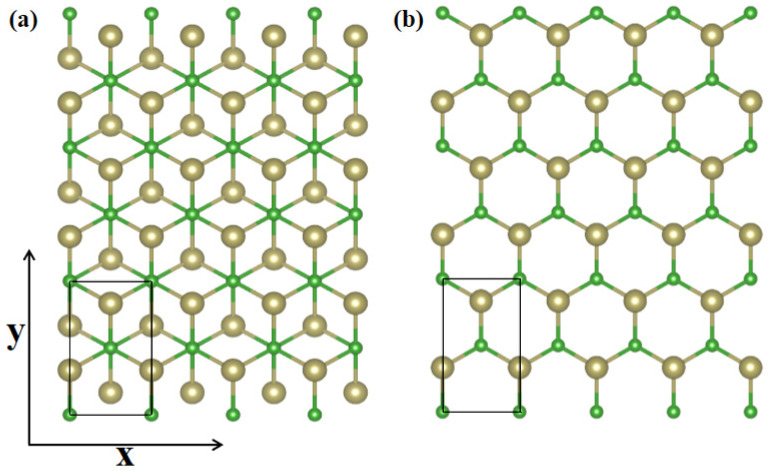
Transformed *α*-TeB (**a**) and *β*-TeB (**b**) structures, the x and y directions are shown on the coordinate axes.

**Figure 5 molecules-26-06404-f005:**
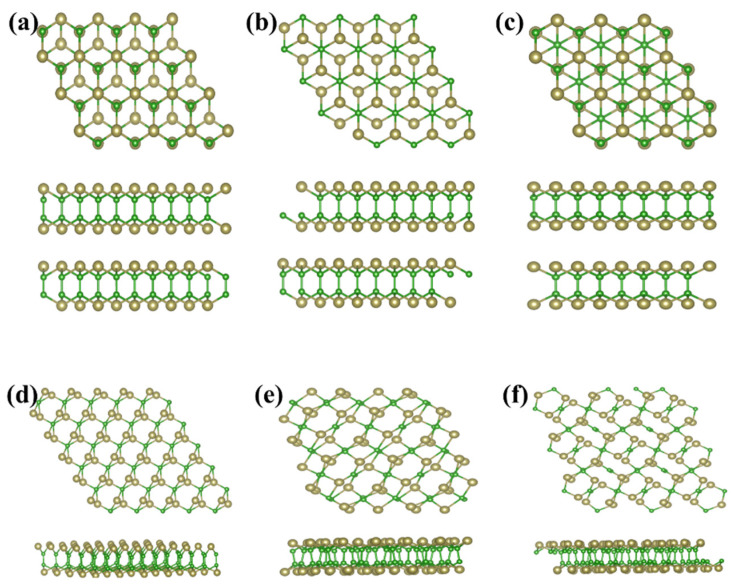
Six structures constructed by stacking(**a–c**) and jointing(**d–f**) *α*-TeB and *β*-TeB.

**Figure 6 molecules-26-06404-f006:**
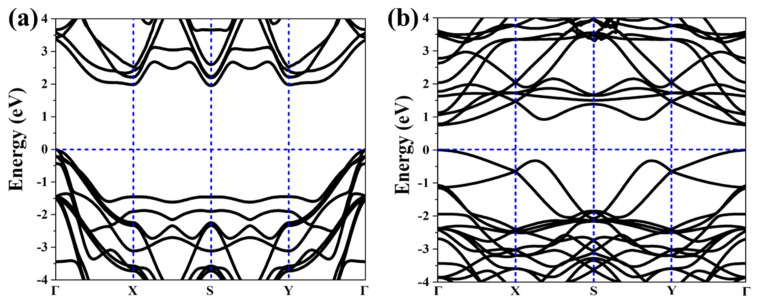
Calculated corresponding band structures of horizonal(**a**) and lateral (**b**) heterostructures with the lowest energy.

**Table 1 molecules-26-06404-t001:** Optimized lattice parameters and atomic positions for *α*-TeB and *β*-TeB.

Phases	Space Group	Lattice Parameters (Å)	Atomic Position
*α*-TeB	*P-3m1*	a = 3.598	Te (0.334, 0.667, 0.562) B (0.000, 1.000, 0.472)
*β*-TeB	*P-6m2*	a = 3.565	Te (0.334, 0.667, 0.425) B (0.667, 0.334, 0.466)

**Table 2 molecules-26-06404-t002:** The calculated elastic constant of the *α*-TeB and *β*-TeB monolayers.

Elastic Constant (N/m)	C_11_	C_12_	C_21_	C_22_	C_66_
*α*-TeB	128.51	19.74	19.74	128.51	54.39
*β*-TeB	142.45	22.17	22.17	142.45	60.14

**Table 3 molecules-26-06404-t003:** Calculated deformation potential constant (EDP), 2D in-plane stiffness (C), effective mass (m*), and carrier mobility (*µ*) along the x and y directions for *α*-TeB and *β*-TeB monolayers at 300 K, respectively.

Phase	Carrier Type	E_DP_ (eV)	C (J·m^−2^)	m* (m_0_)	μ (cm^2^ V^−1^ s^−1^)
*α*-TeB	e(x)	10.63	129.48	0.19	377.91
	h(x)	5.27	129.48	0.61	478.92
	e(y)	5.60	129.21	1.65	45.92
	h(y)	5.14	129.21	0.80	112.41
*β*-TeB	e(x)	12.89	142.84	0.17	414.87
	h(x)	6.60	142.84	0.39	689.79
	e(y)	3.50	143.03	3.65	46.42
	h(y)	6.47	143.03	0.59	84.03

## Data Availability

Data is available from the authors.

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
