# Peer review of "Two-Dimensional TeB Structures with Anisotropic Carrier Mobility and Tunable Bandgap"

_molecules, 2021, doi:10.3390/molecules26216404_

Round 1
Reviewer 1 Report
The manuscript by Zhang et al reports new 2D TeB structures with anisotropic carrier mobility and tubable band gap by using DFT and structure search calculations. I think this is a comprehensive study and the manuscript is well organized. However, I have a minor comment below.
Authors claimed that their stuctures are stable. From the phonon dispersions, imaginary frequencies seem to exist in figure 2a and 2b. For instance, there are flat areas below zero near the gamma point.
I did not find the mobility for specific carrier pockets. For instance, there are several degenerate bands in the conduction bands. Which pocket did the authors calculate the mobility for? There should be a total "average" value for a certain Fermi leverl.
Author Response
Dear Reviewer :
We thank you for the referee reports on our manuscript (Molecules 2021, 26, x). Herein we are sending you a revised version, in which all the comments and suggestions of the Referees have been carefully considered. To save your time and for the Referees’ convenience, here we address the major changes.
Point 1: Authors claimed that their structures are stable. From the phonon dispersions, imaginary frequencies seem to exist in Figure 2a and 2b. For instance, there are flat areas below zero near the gamma point. 

Response 1: First of all, it is well known that two-dimensional layered materials are mostly metastable, imaginary frequencies of some materials will appear near the gamma point, such as MnB2(DOI: 10.1039/c9nr00952c) and Mo2B2(DOI: 10.1039/c8tc06123h), the values of imaginary frequencies of α-TeB and β-TeB monolayers are smaller, can be ignored, therefore, we believe that this point does not affect the subsequent exploration of various properties.
Point 2: I did not find the mobility for specific carrier pockets. For instance, there are several degenerate bands in the conduction bands. Which pocket did the authors calculate the mobility for? There should be a total "average" value for a certain Fermi level.
Response 2: Thanks to reviewers for pointing out this problem, we have made modifications, the VBM and CBM corresponding to the calculated carriers have been reflected in the band structure.
Reviewer 2 Report
In this manuscript, the authors proposed α-TeB and β-TeB monolayers using density functional theory (DFT) combined with the particle swarm-intelligent global structure search method. It is possible to synthesize the two materials in real condition due to the high dynamical and thermal stabilities. Two materials are semiconductors with indirect band gap of about 2 eV, suggesting a promising potential in electronic fields. This work will directly provide some guidance for experiments, and thus I think it can be accepted after a few minor issues are addressed.
1.In several papers published with studying two-dimensional borides, the AIMD results are often discussed with snapshots of the final frame of each molecular dynamics simulation at different temperatures, such as FeB6(DOI: 10.1021/jacs.6b01769) and AlB6 (DOI: 10.1021/jacs.8b13075) published in J. Am. Chem. Soc., the similar discussion style is recommended.
2.During studying the mechanical properties, I suggest the authors to discuss the ideal strength.
3.In Table 3. the typeset should be improved for a better view.
4.In line 221, VBM and CBM appear in paper the first time, authors have better make clear that VBM is short for valence band maximum and CBM is short for conduction band minimum.
5.As the energy change of valence band maximum (VBM) or conduction band minimum (CBM) under strain results the value of carrier mobilities, it is better to give the band decomposed charge densities for VBM and CBM.
6.The authors tune the bandgaps of α-TeB and β-TeB with heterostructures by stacking these two phases, can the bandgap of α-TeB or β-TeB be individually tuned?
7.The authors proposed six heterostructures, only two correspondent band structures are displayed, why?
Author Response
Dear Reviewer :
We thank you for the referee reports on our manuscript (Molecules 2021, 26, x). Herein we are sending you a revised version, in which all the comments and suggestions of the Referees have been carefully considered. To save your time and for the Referees’ convenience, here we address the major changes.
Point 1: In several papers published with studying two-dimensional borides, the AIMD results are often discussed with snapshots of the final frame of each molecular dynamics simulation at different temperatures, such as FeB6(DOI: 10.1021/jacs.6b01769) and AlB6 (DOI: 10.1021/jacs.8b13075) published in J. Am. Chem. Soc., the similar discussion style is recommended. 

Response 1: Thanks to the reviewer for pointing out this problem, discussion of this section has been added to the support information in Figure S1.
Point 2: During studying the mechanical properties, I suggest the authors to discuss the ideal strength.
Response 2: Thanks to the reviewer for pointing out this problem, discussion of this section has been added to the support information in Figure S2.
Point 3: In Table 3. the typeset should be improved for a better view.
Response 3: Thanks to reviewers for pointing out this problem, we have made modifications.
Point 4: In line 221, VBM and CBM appear in paper the first time, authors have better make clear that VBM is short for valence band maximum and CBM is short for conduction band minimum.
Response 4: Thanks to reviewers for pointing out this problem, we have made modifications.
Point 5: As the energy change of valence band maximum (VBM) or conduction band minimum (CBM) under strain results the value of carrier mobilities, it is better to give the band decomposed charge densities for VBM and CBM.
Response 5: Thanks to the reviewer for pointing out this problem, discussion of this section has been added to the support information in Figure S3.
Point 6: The authors tune the bandgaps of α-TeB and β-TeB with heterostructures by stacking these two phases, can the bandgap of α-TeB or β-TeB be individually tuned?
Response 6: Of course, the band gap value can be changed by applying strain, which has been implemented in many works, such as SnSe(DOI: 10.1039/C6TC04692D) and C3N(DOI: 10.1016/j.carbon.2018.12.109). In this paper, we mainly discuss the stacking of these two phases in forming heterojunction.
Point 7: The authors proposed six heterostructures, only two correspondent band structures are displayed, why?
Response 7: The structure with the lowest energy is the most stable. These two correspondent band structures related to the lowest energies atom structures in two stacking mode, so two band structures are representative and are listed.